# Comparative Study of Two Intervention Programmes for Teaching Soccer to School-Age Students

**DOI:** 10.3390/sports7030074

**Published:** 2019-03-26

**Authors:** Juan M. García-Ceberino, Sebastián Feu, Sergio J. Ibáñez

**Affiliations:** Optimization of Training and Sports Performance Research Group (GOERD), Department of Didactics of Music, Plastic and Body Expression, Sports Science Faculty, University of Extremadura, 10004 Cáceres, Spain; sfeu@unex.es (S.F.); sibanez@unex.es (S.J.I.)

**Keywords:** didactic unit, methodology, pedagogical variable, external training load variable

## Abstract

The objective of this study was to design and analyse the differences and/or similarities of two homogeneous intervention programmes (didactic units) based on two different teaching methods, Direct Instruction (DI) and Tactical Games Approach (TGA), for teaching school-age soccer. The sample was composed of 58 tasks, 29 for each intervention programme. The pedagogical and external Training Load (eTL) variables recorded in the Integral System for Training Tasks Analysis (SIATE) were studied. The two intervention programmes were compared using Chi-Square, Mann-Whitney U and the Adjusted Standardized Residuals statistical tests. Likewise, the strength of association of the variables under study was calculated using *Cramer’s Phi* and *Cramer’s V* coefficients. Both intervention programmes had the same number of tasks (*n* = 29), sessions (*n* = 12), game phases (*x^2^*= 0.000; *p* = 1.000), specific contents (*x*^2^ = 5.311; *p* = 0.968) and didactic objectives, as well as different levels of eTL (*U* = 145.000; *p* = 0.000; *d* = 1.357); which are necessary requirements to be considered similar. The differences and/or similarities between both intervention programmes will offer teachers guidelines to develop different didactic units using the specific DI and TGA methodologies.

## 1. Introduction

The teaching–learning process from when an individual begins their sports learning until they are able to put into practice, with a certain amount of effectiveness, what they have learned in an actual game situation, is called sports initiation [1].

Sports initiation can be approached both in the curricular and extracurricular context. Independently of the context in which it takes place, this teaching should be centred on an educational and formative framework [2]. Delgado [3] states that schools are the places where educational and formative sport can be guaranteed, and that it must fulfil a series of characteristics to be considered as such: it must be participative, coeducational, creative, play-oriented, take account of diversity, and promote value education. In this regard, Castejón [4] points to three proposals that every physical education teacher should bear in mind so that the sports teaching has an educational sense: a) a greater incidence on the ambits of cognitive, affective-social and motor knowledge; b) involvement of the students in the design and implementation of the tasks, taking their interests and needs into account; and c) improvement of the evaluation processes, moving towards an educational evaluation.

Sport is part of the curricular content in the area of physical education and, like the rest of curricular contents, requires planning and design using didactic units [5]. Del Valle et al. [6], define didactic units as instruments which facilitate the educational task, permitting the teacher to organise the teaching–learning processes adapted to the group of students. For the correct design, a series of aspects has to be taken into account: the characteristics of the students; an analysis of the context, timing, objectives, contents, methodology, organisation, didactic materials, transversality, curricular adaptation; results and evaluation of the process [7]. These aspects, developed in the didactic units, are put into practice later in the classroom sessions. The classroom session is the minimum unit of programming, based on the didactic unit, so that all together with the rest of the sessions it makes sense for the students’ learning [8].

Invasion sports, among which we find soccer, are those that appear most commonly in teachers’ programmes [9]. The teachers and coaches are those mainly responsible for planning, developing and monitoring sports teaching [10,11]. In this regard, one of the functions of the teacher when planning sports teaching, is to select the method going to be used so that the students acquire the learning contents as efficiently as possible. Two sport teaching approaches stand out: those centred on the teacher (TCA) and those centred on the student (SCA) [12].

Among the TCA, the Direct Instruction (DI) method is the most common. This method is based on the principle of technical competence, a prior requisite before the incorporation of the rules and the game [13]. In addition, the sports teaching is developed in isolated circumstances which are decontextualised from the actual game [14], where the students spend most of the time inactive and with few opportunities for empowerment and creativity [15,16]. The teacher plans, explains and demonstrates the tasks that the students have to perform, and they listen and act according to the determined guidelines; thus, the teacher becomes the protagonist of the teaching–learning process [17,18].

In contrast, the Tactical Games Approach (TGA) stands out among the SCA. In the TGA, an initial form of the game is presented, incorporating tactical problems using small-sided games (SSG) and/or modifying the rules. The practice of technical abilities is introduced later when it is necessary [19,20]. The teacher asks questions of the students during the sports practice so that they resolve the tactical problems set [18,21]. SSG have become a very popular method for all ages and/or levels [22], as they involve the actual movement patterns used in soccer [23] and they are beneficial for the young students in their sports development [24]. The literature presents a great many investigations that study the effects of the SSG formats on physiological, kinematic and technical parameters [25,26,27].

For Rovegno et al. [28], the concepts of technical and tactical ability should not be separated. Although different authors have studied methodologies in sports teaching [29,30,31], few programmes can be found in the literature that have been designed [5], and validated [32] to confirm the efficiency of the different pedagogical methodologies in the teaching of invasion sports in the educational context. In particular, this type of study is unknown in the area of soccer as an invasion sport. The different studies mention the effects provoked by applying intervention programmes according to different specific methodologies [21,33,34]. However, these studies do not mention the planning of such programmes.

Thus, the present study aims to design and analyse the differences and/or similarities of two homogeneous intervention programmes corresponding to two didactic units, Direct Instruction in Soccer (DIS) and Tactical Games Approach to Soccer (TGAS) for teaching soccer in the school context.

## 2. Materials and Methods

### 2.1. Design

This is an instrumental study within the framework of an associative comparative strategy [35], aimed at the design and analysis of two intervention programmes for teaching school-age soccer.

### 2.2. Sample

The study sample was composed of a total of 58 tasks included in two intervention programmes based on two different teaching methods, of which 29 corresponded to the DIS programme and the remaining 29 to the TGAS programme. 

After the programmes had been designed and validated, they were administered to two groups of students from the 5th year of primary education, DIS (*n* = 21) and TGAS (*n* = 20), aged 10 and 11, respectively. The programmes were randomly distributed to the groups. Forty-three percent of the students who participated in the DIS intervention programme and 15% who participated in the TGAS programme practised soccer as an extracurricular activity. All students and teacher were informed about the research protocol, requisites, and benefits, and their written consent was obtained before start of the study. The ethics committee of the University of Extremadura approved the study (nº 09/2018).

### 2.3. Variables

The study variables can be divided into two groups depending on their nature: pedagogical variables and external Training Load (eTL) variables. These variables were recorded in the Integral System for the Analysis of Training Tasks (SIATE by its Spanish acronym) [36].

The pedagogical variables make it possible for the teacher to understand the characteristics of the tasks and facilitate their organisation/structuring. The variables used were: game situation (GS), presence of goalkeeper (POG), game phase (GP), type of content (CONT-G), specific content (CONT-S), teaching means (TM) and level of the opposition (LO) [36]. Each pedagogical variable was structured as a categorical/nominal system of different levels.

The eTL variables used were: degree of opposition (DO), density of the task (DT) percentage of simultaneous performers (PSP), competitive load (CL), game space (GS) and cognitive implication (CI) [36]. Each eTL variable was structured as a categorical/ordinal system with a five-level definition.

The eTL variables make it possible for the teacher to subjectively quantify the load produced by the tasks obtaining a secondary variable: the eTL task load (quantification of eTL). The value of this variable varies between 6 and 30, with four ranges to categorise it: 6–12 (very low), 13–18 (somewhat low), 19–24 (somewhat high) and 25–30 (very high). Similarly, to obtain a more accurate figure for the real load of the task, the eTL was multiplied by the motor time (organisational variable) of the sports activity measured in seconds (eTL × Time) [36].

Table 1 describes the pedagogical and eTL variables recorded in the SIATE [36].

### 2.4. Instruments

The tasks were recorded using the SIATE [36]. This system makes it possible to record and analyse the different parameters that intervene in the process of teaching invasion sports.

The data recorded with the SIATE were then exported to the statistical programme SPSS 21.0 (SPSS. Inc., Chicago, IL, USA), for the descriptive and inferential analysis of the pedagogical and eTL variables present in each intervention programme: DIS and TGAS.

### 2.5. Procedure

This study was divided into two phases: i) The design of the tasks and the drafting of the DIS and TGAS intervention programmes; and ii) A descriptive and inferential analysis to reveal the differences and/or similarities between both intervention programmes.

A series of chronologically ordered actions was used to draft the intervention programmes. First, a literature search was conducted on the DI and TGA methods. Then, the specific contents and didactic objectives to be worked on were established for each of the sessions. These were selected following the proposal by González-Víllora et al. [37]. Then, depending on the methodology, the programme tasks were designed bearing in mind the following elements: number of the task, time, graph, organisation and materials, description of the task, game phase, attacking and/or defensive purpose, teaching means, specific content, game situation and feedback.

Once designed, the tasks were distributed into 12 practical sessions. The sessions did not have the classical structure of a physical education class (warm-up, main part and cool down) [38]. Each session was composed of 4 tasks lasting 10 minutes each, and they were structured progressively, from the simplest (warm-up) to the most complex (culminating activities) [39].

Table 2 shows the distribution of the tasks composing the DIS and TGAS programmes in the 12 practical sessions.

The DIS and TGAS intervention programmes were drawn up with similar structures; thus, they had to have the same number of tasks, sessions, game phases, specific contents and didactic objectives (see Appendix A).

Once the intervention programmes were designed, they were sent to a panel of 13 experts with a recognised trajectory in the study topic to evaluate the suitability of the method and the drafting of the tasks in each programme. Both programmes obtained excellent levels of validity and reliability. Content validity was calculated with Aiken’s V coefficient and its confidence intervals [40], obtaining an exact critical value of 0.69 at a 95% confidence level for the acceptance of the tasks. None of the tasks that made up either intervention programme were eliminated as they all passed the exact established critical value. Cronbach’s *α* coefficient [41], was used to confirm the reliability of the tasks, obtaining a value of 0.97.

After incorporating the improvements suggested by the experts, a descriptive analysis was performed. Finally, the differences and/or similarities between the two intervention programmes were identified using an analysis of their pedagogical and eTL variables. 

### 2.6. Statistical Analysis

The characteristics of the data led to the use of non-parametric mathematical models to test the hypothesis. Firstly, a descriptive analysis was performed to obtain the frequency and the percentage of the categories of each pedagogical and eTL variable present in the DIS and TGAS intervention programmes.

Following this line of analysis, the Adjusted Standardized Residuals (ASR) from the contingency tables [42] were analysed to find the differences between the categories of each pedagogical and eTL variable. The ASR, with a confidence level of 95%, showed the range (ASR > |1.96|) of the categories of each variable differentiating one intervention programme from the other. The categories with residues of > 1.96 indicate that that there are more cases than expected, while those with residues of < −1.96 indicate that there are less cases than expected [43].

Once the descriptive analysis was performed and the ASR studied, an inferential analysis of the different pedagogical and eTL variables was performed to compare both intervention programmes. For this, different statistical tests were used depending on the nature of the variables whether nominal or ordinal [44]. The Chi-Square test (*x*^2^) was applied to the pedagogical variables (nominal) while the Mann-Whitney U test was used for the eTL variables (ordinal), for which effect sizes were calculated using Cohen’s d, according to the following ranges established by Cohen [45]: < 0.000 (adverse), 0.000–0.199 (no effect), 0.200–0.499 (small), 0.500–0.799 (medium) and 0.800– ≥ 1.000 (large).

Lastly, the strength of association was calculated between the study variables. Cramer’s Phi (*Φ_c_*) and Cramer’s V (*V_c_*) were used for the pedagogical variables (nominal × nominal). Cramer’s V (*V_c_*) was also used for the eTL variables (nominal × ordinal) [46]. According to Crewson [47], the strength of association between the variables will depend on the value obtained: <0.100 (small), 0.100–0.299 (low), 0.300–0.499 (moderate) and ≥0.500 (high).

## 3. Results

Table 3 presents the descriptive results and the ASR of the categories of each variable in both intervention programmes.

Figure 1 shows the time in minutes devoted to each of the game phases in both intervention programmes. It can be seen how in spite of the predominance of the attacking phase, there is a balance among the attacking, defensive and mixed phases. This process will offer the students balanced and complete training [48].

The descriptive results and ASR of the categories of each eTL variable in both intervention programmes are presented in Table 4.

The mean quantification in eTL and eTL × Time of the tasks for each intervention programme is shown in Table 5.

Table 6 presents the differences and/or similarities between both intervention programmes using the Chi-Square statistical test for the pedagogical variables. 

Lastly, the differences and/or similarities between both intervention programmes using the Mann-Whitney U test for the eTL variables are shown in Table 7.

## 4. Discussion

The purpose of this study was to design and analyse the differences and/or similarities between two homogeneous intervention programmes (didactic units): DIS and TGAS, based on two different teaching methods: DI and TGA respectively. The two intervention programmes are valid and reliable for teaching soccer in the school context, as well as for comparing the efficiency of the two teaching methodologies in the learning of soccer in primary education. The results show significant differences in some of the pedagogical and eTL variables between the two intervention programmes, due to the particularity of each methodology. The main advantages of the TGA over DI can be found in the greater participation and motivation of the students during the physical education classes and the fact that they acquire a greater understanding and retention of the learning due to the use of games. 

The pedagogical organisation of the tasks influences the selected teaching method. Currently, no studies have been found that have discriminated in which parameters one or other of the methodologies is located. The results of this study show significant differences (*p* < 0.05) in the following variables: game situation, type of contents, teaching means and level of opposition between the two intervention programmes. The DIS intervention programme seeks the achievement of technical moves using analytical tasks, incorporating the global tasks at the end of the teaching process [12,49,50]. In contrast, the TGAS intervention programme aims at the resolution of tactical problems using play tasks or situations from the real competitive game where the students have to make decisions [19,51,52]. The results obtained coincide with those of the study aimed at designing two homogeneous intervention programmes, Direct Instruction in Basketball (DIB) and Tactical Game in Basketball (TGB), for teaching basketball in primary education [5]. However, there were no significant differences (*p* > 0.05) between both programmes in the following variables: game phase and specific content. These results show that the design of the tasks for each programme was similar, although based on different methodologies. González-Víllora et al. [53], indicate that the students learn the elements of attack before those of defence. Thus, the two intervention programmes gave priority to the attack phase over the defence phase. Similarly, there is a balance between the phases of attack, defence and mixed fomenting balanced and complete training for the students [48]. Different programmes drawn up for teaching soccer from a vertical perspective also give priority to learning attack over defence [54].

Regarding the eTL variables, the results show significant differences between the DIS and TGAS programmes in the following variables: degree of opposition, density of the task, competitive load and cognitive implication. These variables show that the quantification of eTL is higher in the TGAS (*M* = 20.21; moderately high) and lower in the DIS intervention programmes (*M* = 14.34; moderately low), coinciding with the results obtained by Chen et al. [55], who indicate that sports practice in constructivist methodologies provokes higher levels of intensity. The quantification of eTL can be affected by the modification of the structural and formal parameters of the tasks: the rules of the game, the dimensions of the space, the number of players involved, the goal count, the duration of the task etc. [22,56]. Different authors have focused on studying the dimensions of the space [57,58] and the number of players involved [59,60]. In such a way that smaller spaces and fewer players provoke higher levels of eTL as a consequence of the greater contact among the players and with the ball [61]. These types of tasks are called small-sided games, SSG [62], and they are especially suitable for sports development in the young [24]. The results obtained show significant differences in eTL and eTL*Time between both intervention programmes. These results coincide with a study to design homogeneous intervention programmes DIB and TGB, for teaching basketball in primary education [5], or for contrasting the load between training and actual competition in a women’s basketball team in the sphere of sports training [63]. However, no differences existed in the percentage of simultaneous participants and play space variables.

The choice of one programme or another for teaching soccer depends on the knowledge and methodological approach of the teacher [64]. However, the TGAS intervention programme appears to be the most favourable programme due to the fact that this methodology offers a wide range of motor experiences, as well as developing different abilities, capacities, skills and competences suitable for the psychoevolutionary characteristics of the students [65]. Similarly, Chatzipanteli et al. [66] state that the TGA could improve the metacognitive behaviour of the students in physical education classes in primary education. This method works on technical, tactical and physical aspects using play, SSG, thus obtaining a better retention of learning [67]. In contrast, in the DI method, the students acquire little understanding of the game during the physical education lessons, and as a result their decision making abilities are deficient [68]. Furthermore, the DI method makes it difficult to keep up the motivation and performance of the students during long periods of time, as the situations proposed are not very stimulating because they do not possess the essential aspects of the game. [69,70]. The participation and motivation of the students in physical education is greater with the TGA because of the fun and enjoyment of the game [71]. The use of TGA by the teachers can also help the students to attain adequate levels of physical activity for health in the physical education classes; however, the prevalence of the DI method in this subject is the cause of students’ high levels of inactivity [15,72,73]. Thus, different authors propose the TGA for school physical education, and they are used in isolated cases especially by teachers who show a preference for these methods [51,74]. The TGA are used to a lesser extent due to the difficulty in implementing them because of their emphasis on understanding the game [19,75] and the lack of information [4].

Lastly, it is important to indicate the very small number of intervention programmes designed according to a specific teaching–learning method [5], and validated by a panel of experts [32] for teaching invasion sports in the school context. Specifically, this type of study is unknown in the sport of soccer. The different studies found in the literature point to the effects produced by the application of intervention programmes using specific methodologies [21,33,34], but do not mention how such programmes were designed and validated. The results obtained in the validation of the DIS and TGAS intervention programmes show excellent internal consistency and reliability, which makes them recommendable for use in the school context and in the context of sports training. For this reason, this study has the strong point of offering teachers, coaches and researchers, guidelines that can serve as a reference for drawing up programmes (didactic units) according to the specific methodologies of DI and TGA. One limitation to consider is the duration of the intervention programmes, 10–12 sessions, which depends on the duration of the didactic units in the education system. Thus, these intervention programmes have shorter duration than extracurricular sport intervention programmes. Before implementing the intervention programmes, it is necessary to adapt them to the level of the students because they are constantly exposed to soccer. Playing soccer in the extracurricular context can influence the level of learning achieved after the intervention programmes.

## 5. Conclusions

The DIS and TGAS intervention programmes can be used by physical education teachers for teaching soccer in schools. The choice of one programme or another will depend on the knowledge, methodological approach and life experience of the teacher.

The study of the differences and/or similarities of both intervention programmes will offer the teachers guidelines for drawing up programmes (didactic units) according to the specific methodologies of DI and TGA. In this regard, TGA is more favourable than DI due to the use of games, SSG, which provokes greater participation and motivation on the part of the students during the physical education classes, as well as a better understanding and retention of the learned contents. 

These programmes will also make it possible for researchers to analyse the level of learning acquired by the students after their application, as well as contrasting the effects of the different teaching methodologies in the learning of school soccer. 

## Figures and Tables

**Figure 1 sports-07-00074-f001:**
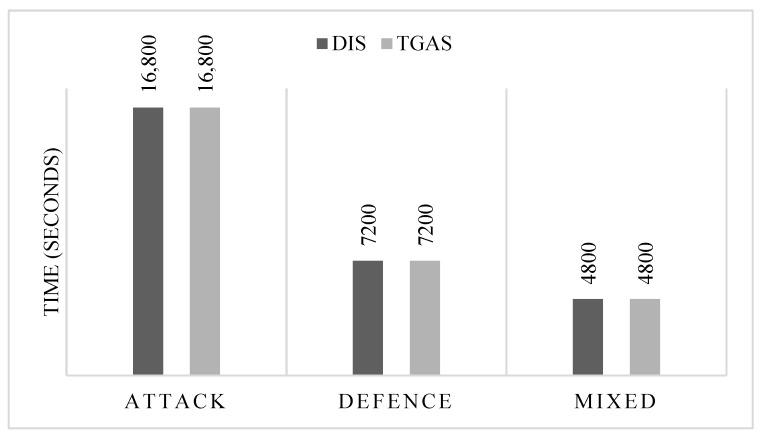
Time devoted to each game phase in both intervention programmes.

**Table 1 sports-07-00074-t001:** Synthesis of the pedagogical and eTL variables.

**Pedagogical Variables **	**Description**
GS	Groups of players that the teachers and coaches design for each of the tasks (e.g. 2 × 1; 2 being the number of attackers and 1 the number of defenders).
POG	Presence of goalkeeper in the task.
GP	Game phase on which the task objective is mainly focused.
CONT-G	The contents (attack and defence) are grouped in individual, group and team contents, as well as tactical behaviours and technical moves.
CONT-S	Specific contents for each sports discipline.
TM	Sports motor activities that serve to develop technical and tactical contents.
LO	Level of opposition in the task design.
**eTL Variables**	**Description**
DO	Degree of opposition based on the number of opponents in the task.
DT	Indicates subjectively the intensity with which the task is developed.
PSP	Indicates the level of participation of the players during the task.
CL	Refers to the emotional and psychological load that the players bear when they have to carry out a task under pressure to achieve a result.
GS	The space in which the players have to carry out the proposed tasks.
CI	Refers to the tactical load, i.e. the attention that the player has to give to team mates and opponents.
eTL task load	Obtained by adding the value assigned to each of the six eTL variables (1 to 5 points). GO + DT + PES + CC + EJ + IC=quantification of eTL.
eTL × Time	Calculated by multiplying eTL by the useful time that the players have been practising measured in seconds.

Note: GS = Game situation; POG = Presence of goalkeeper; GP = Game phase; CONT-G = Type of contents; CONT-S = Specific content; TM = Teaching means; LO = Level of opposition.

**Table 2 sports-07-00074-t002:** Distribution of the tasks composing the DIS and TGAS programmes.

**DIS**											
**S1**	**S2**	**S3**	**S4**	**S5**	**S6**	**S7**	**S8**	**S9**	**S10**	**S11**	**S12**
DI10	DI8	DI10	DI12	DI15	DI16	DI19	DI24	DI26	DI26	DI28	DI29
DI3	DI7	DI9	DI11	DI14	DI15	DI18	DI23	DI25	DI25	DI27	DI28
DI2	DI6	DI8	DI8	DI13	DI12	DI17	DI22	DI24	DI23	DI20	DI27
DI1	DI5	DI2	DI5	DI5	DI11	DI16	DI21	DI21	DI22	DI4	DI20
**TGAS**											
**S1**	**S2**	**S3**	**S4**	**S5**	**S6**	**S7**	**S8**	**S9**	**S10**	**S11**	**S12**
TG1	TG5	TG2	TG6	TG6	TG11	TG4	TG21	TG23	TG22	TG19	TG20
TG2	TG6	TG8	TG7	TG13	TG12	TG17	TG22	TG24	TG23	TG20	TG27
TG3	TG7	TG9	TG11	TG14	TG13	TG18	TG23	TG25	TG25	TG27	TG28
TG4	TG8	TG10	TG12	TG15	TG16	TG19	TG24	TG26	TG26	TG28	TG29

**Table 3 sports-07-00074-t003:** Differences and/or similarities between the pedagogical variables of the DIS and TGAS programmes.

**Variable**	**Category**	**DIS**				**TGAS**			
		***n***	**%**	**ASR**		***n***	**%**	**ASR**	
GS	0 vs. 1	1	3.4	1.0		0	0	−1.0	
	1 vs. 0	8	27.6	3.0	*	0	0	−3.0	*
	1 vs. 1	6	20.7	−1.2		10	34.5	1.2	
	2 vs. 0	7	24.1	2.8	*	0	0	−2.8	*
	2 vs. 1	2	6.9	−0.5		3	10.3	0.5	
	2 vs. 2	1	3.4	0.0		1	3.4	0.0	
	3 vs. 0	1	3.4	1.0		0	0	−1.0	
	3 vs. 1	0	0	−1.0		1	3.4	1.0	
	3 vs. 2	0	0	−1.8		3	10.3	1.8	
	4 vs. 2	0	0	−1.0		1	3.4	1.0	
	4 vs. 4	0	0	−1.0		1	3.4	1.0	
	5 vs. 1	0	0	−1.0		1	3.4	1.0	
	5 vs. 4	0	0	−1.8		3	10.3	1.8	
	5 vs. 5	1	3.4	0.0		1	3.4	0.0	
	N vs. N	1	3.4	0.0		1	3.4	0.0	
	1 vs. Large group	1	3.4	−0.6		2	6.9	0.6	
	Combined situation	0	0	−1.0		1	3.4	1.0	
POG	With goalkeeper	1	3.4	1.0		0	0	−1.0	
	Without goalkeeper	28	96.6	−1.0		29	100	1.0	
GP	Attack	19	65.5	0.0		19	65.5	0.0	
	Defence	6	20.7	0.0		6	20.7	0.0	
	Mixed	4	13.8	0.0		4	13.8	0.0	
CONT-G	AITTB	1	3.4	−3.2	*	11	37.9	3.2	*
	DITTB	0	0	−2.1	*	4	13.8	2.1	*
	AITTM	10	34.5	3.5	*	0	0	−3.5	*
	DITTM	6	20.7	2.6	*	0	0	−2.6	*
	AGTTB	3	10.3	−1.4		7	24.1	1.4	
	DGTTB	0	0	−1.4		2	6.9	1.4	
	AGTTM	8	27.6	3.0	*	0	0	−3.0	*
	ATTTB	1	3.4	−1.7		5	17.2	1.7	
CONT-G 2 ^1^	CTTID	2	6.9	1.6		0	0	−1.6	
	CTTGD	1	3.4	−1.4		3	10.3	1.4	
	CTTED	1	3.4	0.0		1	3.4	0.0	
CONT-S	Pass-control	5	17.2	0.4		4	13.8	−0.4	
	Ball control: progression	2	6.9	−0.5		3	10.3	0.5	
	Ball control: protection	2	6.9	0.0		2	6.9	0.0	
	Progression with support	1	3.4	0.0		1	3.4	0.0	
	Shot at goal	2	6.9	1.4		0	0	−1.4	
	Progression to shoot at goal	1	3.4	−1.0		3	10.3	1.0	
	Progression through passes to score at goal	3	10.3	0.0		3	10.3	0.0	
	Dribbling on the run	2	6.9	0.0		2	6.9	0.0	
	Dribbling past opponent to shoot	1	3.4	0.0		1	3.4	0.0	
	Interception: Shot at goal or approach	2	6.9	0.0		2	6.9	0.0	
	Interception: passes between opponents	4	13.8	0.0		4	13.8	0.0	
	Situations played: attack and defence	4	13.8	0.0		4	13.8	0.0	
TM	SPE	12	41.4	3.9	*	0	0	−3.9	*
	CPE	5	17.2	2.3	*	0	0	−2.3	*
	SNSG	1	3.4	−0.6		2	6.9	0.6	
	SSG	10	34.5	−1.6		16	55.2	1.6	
	CSG	0	0	−3.5	*	10	34.5	3.5	*
	Adapted sport/SSG	1	3.4	0.0		1	3.4	0.0	
LO	Without opposition	10	34.5	3.0	*	1	3.4	−3.0	*
	Static obstacles	7	24.1	2.8	*	0	0	−2.8	*
	Dynamic obstacles	2	6.9	1.4		0	0	−1.4	
	With opposition	10	34.5	−5.0	*	28	96.6	5.0	*

Note: GS = Game situation; POG = Presence of goalkeeper; GP = Game phase; CONT-G = Type of content; CONT-S = Specific content; TM = Teaching means; LO = Level of the opposition. AITTB = Attacking individual technical-tactical behaviours; DITTB = Defensive individual technical-tactical behaviours; AITTM = Attacking individual technical-tactical moves; DITTM = Defensive individual technical-tactical moves; AGTTB = Attacking group technical-tactical behaviours; DGTTB = Defensive group technical-tactical behaviours; AGTTM = Attacking group technical-tactical moves; ATTTB = Attacking team technical-tactical behaviours; DTTTB = Defensive team technical-tactical behaviours. SPE = Simple practical exercise; CPE = Complex practical exercise; SNSG = Simple non-specific game; SSG = Simple specific game; CSG = Complex specific game. ^1^ This variable was added to be able to indicate the type of contents when working on a mixed game phase. * ASR > |1.96|.

**Table 4 sports-07-00074-t004:** Differences and /or similarities in the eTL between the DIS and TGAS programmes.

Variable	Category	DIS				TGAS			
		*n*	*%*	ASR		*n*	*%*	ASR	
DO	Without opposition	18	62.1	5.1	*	0	0	−5.1	*
	Numerical superiority of 3 or + students	1	3.4	−1.0		3	10.3	1.0	
	Numerical superiority of 2 students	0	0	−1.4		2	6.9	1.4	
	Numerical superiority of 1 student	2	6.9	−2.3	*	9	31.0	2.3	*
	Numerical equality	8	27.6	−1.9		15	51.7	1.9	
DT	Walking	5	17.2	2.3	*	0	0	−2.3	*
	Gentle pace	13	44.8	4.1	*	0	0	−4.1	*
	Intensity with rest	8	27.6	−1.9		15	51.7	1.9	
	Intensity without rest	2	6.9	−3.1	*	12	41.4	3.1	*
	High intensity without rest	1	3.4	−0.6		2	6.9	0.6	
PSP	<20%	10	34.5	1.5		5	17.2	−1.5	
	21–40%	4	13.8	−0.4		5	17.2	0.4	
	41–60%	0	0	−1.0		1	3.4	1.0	
	61–80%	0	0	-		0	0	-	
	>81%	15	51.7	−0.8		18	62.1	0.8	
CL	Activity in technical moves	19	65.5	5.3	*	0	0	−5.3	*
	Opposition not counted	9	31.0	−3.2	*	21	72.4	3.2	*
	Opposition counted	0	0	−2.8	*	7	24.1	2.8	*
	Matches of all kinds	1	3.4	0.0		1	3.4	0.0	
GS	Static activity	5	17.2	1.7		1	3.4	−1.7	
	Small spaces	16	55.2	0.0		16	55.2	0.0	
	Medium spaces	7	24.1	−1.1		11	37.9	1.1	
	Large spaces	1	3.4	1.0		0	0	−1.0	
	Repetition of spaces	0	0	−1.0		1	3.4	1.0	
CI	Individual intervention	10	34.5	3.5	*	0	0	−3.5	*
	Intervention of 2 students	16	55.2	0.0		16	55.2	0.0	
	Intervention of 3 students	1	3.4	−1.7		5	17.2	1.7	
	Intervention of 4 students	0	0	−1.4		2	6.9	1.4	
	Intervention of all the students	2	6.9	−1.5		6	20.7	1.5	

Note: DO = Degree of opposition; DT = Density of the task; PSP = Percentage of simultaneous performers; CL = Competitive load; GS = Game space; CI = Cognitive implication. *ASR > |1.96|.

**Table 5 sports-07-00074-t005:** Mean quantification of eTL and eTL × Time of the tasks for each programme.

Variable	IP	*M ± DT*	*min*	*max*
eTL	DIS	14.34 ± 5.01	8	28
	TGAS	20.21 ± 3.72	16	28
eTL × Time	DIS	6885.52 ± 2404.37	3840.00	13,440.00
	TGAS	9699.31 ± 1783.95	7680.00	13,440.00

**Table 6 sports-07-00074-t006:** Relation and degree of association between the pedagogical variables.

Variable	IP	*M ± DT*	*x* ^2^	*gl*	*p*		*V_C_*	*p*	
GS	DIS	27.25 ± 19.15	29.533	16	0.021	*	0.714	0.021	*
	TGAS	38.25 ± 14.34							
POG^1^	DIS	1.96 ± 1.86	1.018	1	0.313		0.132	0.313	
	TGAS	2.00 ± 0.00							
GP	DIS	1.48 ± 0.74	0.000	2	1.000		0.000	1.000	
	TGAS	1.48 ± 0.74							
CONT-G	DIS	5.00 ± 3.27	42.600	7	0.000	*	0.857	0.000	*
	TGAS	6.00 ± 2.00							
CONT-G 2^2^	DIS	5.00 ± 3.83	3.000	2	0.223		0.612	0.223	
	TGAS	7.00 ± 2.00							
CONT-S^3^	DIS	-	5.311	13	0.968		0.303	0.968	
	TGAS	-							
TM	DIS	4.75 ± 1.50	28.718	5	0.000	*	0.704	0.000	*
	TGAS	5.75 ± 1.26							
LO	DIS	2.76 ± 1.74	24.890	3	0.000	*	0.655	0.000	*
	TGAS	4.86 ± 0.74							

Note: GS = Game situation; POG = Presence of goalkeeper; GP = Game phase; CONT-G = Type of content; CONT-S = Specific content; TM = Teaching means; LO = Level of the opposition. ^1^ The Φ coefficient was used for this variable instead of V. ^2^ This variables was added to be able to indicate the type of contents when working on a mixed game phase. ^3^ The mean and standard deviation of this chain variable cannot be calculated as the values cannot be admitted. * *p* < 0.05.

**Table 7 sports-07-00074-t007:** Relation and degree of association between the variables for eTL.

Variable	IP	*M ± DT*	*U*	*p*		*d*	*V_C_*	*p*	
DO	DIS	2.34 ± 1.84	192.500	0.000	*	1.052	0.690	0.000	*
	TGAS	4.24 ± 0.99							
DT	DIS	2.34 ± 0.97	130.000	0.000	*	1.474	0.690	0.000	*
	TGAS	3.55 ± 0.63							
PSP	DIS	3.21 ± 1.92	355.000	0.253		0.270	0.229	0.384	
	TGAS	3.72 ± 1.71							
CL	DIS	2.41 ± 0.68	123.000	0.000	*	1.530	0.729	0.000	*
	TGAS	3.31 ± 0.54							
GS	DIS	2.14 ± 0.74	332.000	0.124		0.367	0.309	0.235	
	TGAS	2.45 ± 0.74							
CI	DIS	1.90 ± 1.01	198.500	0.000	*	1.017	0.536	0.002	*
	TGAS	2.93 ± 1.22							
eTL	DIS	14.34 ± 5.01	145.500	0.000	*	1.357	0.793	0.004	*
	TGAS	20.21 ± 3.72							
eTL × Time	DIS	6885.52 ± 2404.37	145.500	0.000	*	1.357	0.793	0.004	*
	TGAS	9699.31 ± 1783.95							

Note: DO = Degree of opposition; DT = Density of the task; PSP = Percentage of simultaneous performers; CL = Competitive load; GS = Game space; CI = Cognitive implication. * *p* < 0.05.

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
