# Peer review of "Comparative Study of Two Intervention Programmes for Teaching Soccer to School-Age Students"

_sports, 2019, doi:10.3390/sports7030074_

Round 1

Reviewer 1 Report

General comments

The article requires a major grammatical revision.

Where a sentence is started with a bracketed reference, please spell out ‘Author, year’ e.g. Smith (2000), ………………

I think more reference could be made to SSG in the Introduction, as there is a substantial amount of research in this area.

I think the tables in the Results section could be made easier to follow with descriptions of the abbreviated variables as a note under the tables, so that the reader doesn’t have to go back to find them.

Are there specific definitions for the variables in Cont-S? the interpretation of these should be included to ensure study replication.

There are numerous tables of results, could it be made simpler to read. I’m not sure how myself, but just a thought which will keep the reader engaged.  

The graph should be revised to a publishable standard with correct labelling and colour (blacks, greys and whites). In fact, the graph might not be necessary as there is no difference within attack, defence or mixed which could be included in text.

‘teacher having life experience’ does this mean that the teacher needs to understand the level of skill of the students before planning the teaching lesson. If so I think you need to make this more explicit. Teacher life experience is vague.

I believe the football world has moved to using mostly SSG/TGA and therefore makes the novelty of the study questionable. I think a focus of the Discussion should be to discuss the benefit of TGA over DI, as it is potentially a better method of learning and for retention of learning.

Author Response

Correcciones:

- First, a native translator performed a grammatical revision of the manuscript.

- The citations were adapted to the style of the journal. In addition, more English references were added.

- Abstract. In the results section, some statistical data (n; p; d) were added.

- Introduction. More studies that investigated the Small-Sided Games (SSG) were cited.

- Method. The characteristics of the students who participated in the intervention programmes designed in this study were mentioned. In addition, the reference of the ethics commitee was added. Likewise, a table to describe the pedagogical and eTL variables was created.

- Results. Each table was added a note to describe the abbreviated pedagogical and eTL variables. Thus, the reader is facilitated reading the manuscript. In figure 1 a black and white scale was used, and its quality was improved.

- Discussion. The benefit of the Tactical Games Approach (TGA) versus the Direct Instruction (DI) was discussed. In the last paragraph, the strengths and limitations of the study were added.

- Conclusion. It was indicated which methodology, TGA or DI, is more favorable for the teaching of school soccer.

Reviewer 2 Report

I commend the authors for their study.

The aim of this study was to design and analyse the differences and/or similarities of two homogeneous intervention programmes (didactic units) based on two different teaching methods, Direct Instruction (DI) and Tactical Games Approach (TGA), for teaching school-age soccer. Overall, this study provides an interesting sport teaching approach, which might be useful for sport scientists and soccer coaches and practitioners. Nevertheless, some general concerns should be highlighted. The results should be better described. Describe the main results of each table.

Specific comments:

- Line 30: Please change the citation “[3]” to the style format “authors [number]”.

- Line 33: Please change the citation “[4]” to the style format “authors [number]”. When appropriate, correct it along the text (e.g., line 39 “[6]”; line 67 “[19]”; and others places).

- Line 81 “Subjects”: How many students participated in the study? Authors need to clarify the characteristics of the students (e.g., age, experience with “deliberate game” and/or “deliberate practice”).

- Line 85 “Variables”: I encourage the authors to create a Table describing better the pedagogical and external variables (as well as the ways of calculating them). This will help readers who are unaware of this method.

- Line 109 “Procedures”: This section should be better in another place; after “Design”.

- Line 169 “Figure 1”: I recommend authors create the Figure 1 in a different software (e.g. Origin, Graphpad).

- Line 183: Please, remove this sentence “However, we do not know in what parameters the two programmes differ or are similar”. In addition, restructure this sentence “In this regard, the results show significant differences in some of the pedagogical and eTL variables between the two intervention programmes, due to the particularity of each methodology”. What are the main advantages of one approach over the other? Show in general the main advantages in the first paragraph of the discussion.

- This study is not without limitations. Please, describe the limitations in the last paragraph of the discussion section.

- During the discussion, the authors recommend the use of the TGAS approach: “…the TGAS intervention programme appears to be the most favourable programme due to the fact that this methodology offers a wide range of motor experiences, as well as developing different abilities, capacities, skills and competences suitable for the psychoevolutionary characteristics of the students”. However, in the conclusion section the authors do not position themselves (according to the results presented) on which approach has the greatest advantage. Please be clearer.

Author Response

(The authors gave the same response as above.)

Reviewer 3 Report

General comment

This is an interesting study on pedagogical aspects of soccer in a school context. Considering its originality and practical applications, I would recommend it for publication once the authors address a few issues. The most serious issue is the extended use of non-English literature which makes it impossible for all non-Spanish scientists to control the sources of the study. Thus, I would suggest that the authors substitute all non-English literature with English one in order to publish their paper in a high-quality international journal.

Specific comments

Abstract, l. 15-18: Delete “The two intervention... coefficients”.

Abstract, l. 18-19: Develop the results section in 2-3 sentences presenting some numbers (e.g. mean+/-SD, p values, effect size)...

Figure 1. The quality is poor. You should use better software and black & white.

Discussion, l.237. A paragraph in needed before conclusions presenting limitations, strength and practical applications for this study.

Conclusions. Small paragraphs should be integrated into larger here and throughout the text.

Author Response

(The authors gave the same response as above.)
